## [Editor Report · Decision Letter 0]

18 Jun 2025

Library Size-Stabilized Metacells Construction Enhances Co-Expression Network Analysis in Single-Cell Data

PLOS Computational Biology

Dear Dr. Zhu,

Thank you for submitting your manuscript to PLOS Computational Biology. After careful consideration, we feel that it has merit but does not fully meet PLOS Computational Biology's publication criteria as it currently stands. Therefore, we invite you to submit a revised version of the manuscript that addresses the points raised during the editorial assessment process.

Please submit your revised manuscript within 60 days Aug 18 2025 11:59PM. If you will need more time than this to complete your revisions, please reply to this message or contact the journal office at ploscompbiol@plos.org. Please include the following items when submitting your revised manuscript:

* A rebuttal letter that responds to each point raised by the editor. You should upload this letter as a separate file labeled 'Response to Reviewers'. This file does not need to include responses to formatting updates and technical items listed in the 'Journal Requirements' section below.

We look forward to receiving your revised manuscript.

Kind regards,

Marcel Holger Schulz, Ph.D.

Academic Editor

PLOS Computational Biology

Ilya Ioshikhes

Section Editor

PLOS Computational Biology

**Additional Editor Comments :**

I have read your paper and there are a few things that I would like you to address before I can sent it out for peer-review:

1. Cite and/or compare your approach for construction of metacells with existing methods that have been designed for that purpose using the same benchmarking you do already in the paper:

- MetaCell: analysis of single-cell RNA-seq data using K-nn graph partitions. Baran et al. Gen Biol 2019

- Building and analyzing metacells in single-cell genomics data. Bilous et al. Mol Sys Biol 2024

- MetaQ: fast, scalable and accurate metacell inference via single-cell quantization. Li et al. Nature Communciations 2025

2. Your argumentation is that the library size of the metacells is an important factor to control in the creation of metacells. In your approach LSMetacell you do this as described in Algorithm 1. In order to prove your argument on the level of the benchmark that you use, it would be useful to create an alternative implementation of LSMetacell, where you do not control for library size at the time of aggregation, but simply control for the number of metacells for example. Then you should check if the module preservation is negatively affected by leaving out library size control.

**Journal Requirements:**

At this stage, the following Authors/Authors require contributions: Tianjiao Zhang. Please ensure that the full contributions of each author are acknowledged in the "Add/Edit/Remove Authors" section of our submission form.

2) We note that your Manuscript files are duplicated on your submission. Please remove any unnecessary files from your revision, and make sure that only those relevant to the current version of the manuscript are included.

3) If any authors received a salary from any of your funders, please state which authors and which funders.

6) Thank you for stating "The authors declare no competing interests." If you have no competing interests to declare, please state "The authors have declared that no competing interests exist."

**Figure resubmission:**

**Reproducibility:**



---

## [Decision Letter · Decision Letter 1]

17 Aug 2025

Library Size-Stabilized Metacells Construction Enhances Co-Expression Network Analysis in Single-Cell Data

PLOS Computational Biology

Dear Dr. Zhu,

Thank you for submitting your manuscript to PLOS Computational Biology. After careful consideration, we feel that it has merit but does not fully meet PLOS Computational Biology's publication criteria as it currently stands. Therefore, we invite you to submit a revised version of the manuscript that addresses the points raised during the review process.

Please submit your revised manuscript within 60 days Oct 17 2025 11:59PM. If you will need more time than this to complete your revisions, please reply to this message or contact the journal office at ploscompbiol@plos.org. Please include the following items when submitting your revised manuscript:

We look forward to receiving your revised manuscript.

Kind regards,

Marcel Holger Schulz, Ph.D.

Academic Editor

PLOS Computational Biology

Ilya Ioshikhes

Section Editor

PLOS Computational Biology

**Journal Requirements:**

2) Please amend your detailed Financial Disclosure statement. This is published with the article. It must therefore be completed in full sentences and contain the exact wording you wish to be published.

**Reviewers' comments:**

Reviewer's Responses to Questions

Reviewer #1: In the context of scRNA-seq, the authors address the well-known issue that normalizing library sizes through linear scaling introduces bias into estimated gene-gene correlations across single cells. As a remedy, they propose combining meta-cell formation with explicit control for library size variation. Specifically, they introduce a heuristic greedy algorithm that aggregates similar cells into meta-cells until the total count within each meta-cell reaches a predefined target library size. Through extensive validation experiments, they demonstrate that this approach effectively reduces bias in gene-gene correlation estimates, while preserving biological signal.

The idea is simple yet convincing. The validation experiments are generally convincing. Nevertheless, I would like to raise several points that could help further improve the manuscript.

Major comments:

The authors emphasize that their method controls for library size, unlike other approaches. However, it is important to acknowledge that standard library size normalization by scaling also “controls” library size, just in a different way. In my view, the key distinction lies in how variability across genes is handled. Scaling adjusts all gene counts within a (meta-)cell by the same factor. When different cells have different scaling factors, this uniform scaling across genes artificially inflates gene-gene correlations. In contrast, meta-cell formation aggregates counts from multiple cells, preserving more gene-specific variability: some genes may increase their counts by, say, 10%, while others only by 5%. This heterogeneity reduces (or even eliminates) spurious correlations while still approximately equalizing library sizes across meta-cells. I suggest that the authors make this conceptual point more explicit in the manuscript.

I believe it is also important to note that the generated meta-cell data cannot fully eliminate false correlations, as the data remain approximately compositional. One would still expect to observe the characteristic negative correlations across genes that arise in compositional data: if one gene is more highly expressed in a meta-cell, other genes must have correspondingly lower counts to maintain the fixed total library size. Were such negative correlations still observed in the meta-cell data?

1. Figure 1: The main conclusion, that increasing variance in library size leads to inflated gene-gene correlations, is clearly supported. However, I am curious why the authors chose to present the bias as a function of mean gene expression, and how they interpret the specific shape of the resulting curves. Furthermore, I would appreciate an explanation for the small but consistent positive bias observed even under constant library sizes. What is the origin of this residual bias?

2. Figure 2: The authors write that “LSMetacell demonstrated higher and stable module preservation.” I believe a more nuanced discussion is needed here, as this claim does not appear to hold uniformly across all cell types shown in the figure.

The manuscript refers to a dual “optimization” strategy underlying their algorithm. Since no formal optimization problem is actually solved, I recommend rewording this to avoid overstating the methodological rigor.

Minor comments:

On line 179, the authors write: “We found that the coefficient of variation of meta-cells obtained by different methods was in the order of hdWGCNA > metaQ > primary > metacell2 > LSMetacell should be interpreted...” Although this is part of the results section, a brief comment on the scientific significance of this observation would be helpful for the reader.

In Algorithm 1, I believe that the $x'$ in the denominator of the formula on line 21 should be $w_{x'}$.

Reviewer #2: Review: Library Size-Stabilized Metacells Construction Enhances Co-Expression Network

Analysis in Single-Cell Data

While multiple methods have been developed to build metacells, few methods have explored the influence of variations in library sizes before and after metacell construction. This paper presents a tool called LSMetacell to build metacells while stabilizing the library sizes across metacells. This method has been benchmarked against other tools designed to build metacells: hdWGCNA, MC2 and MetaQ as well as a version of LSMetacell which does not consider library size in the metacells building step. While LSMetacells is considering an important bias in (sc)RNA-Seq analyses, further analyses need to be added to evaluate in a more comprehensive manner the quality of metacells built with LSMetacells and the performance of this tool against standard approaches.

Major comments:

1. The authors generated null datasets in which genes are not co-expressed and evaluated the level of gene co-expression at the single-cell level, for different library size variations across cells. The authors should show that their method performs well on the null datasets, i.e. minimal biases should be observed after building metacells on the null datasets.

2. Also, the analysis based on the null datasets shows an increase in averaged gene correlations when variations in library sizes increase. However, the levels of correlations are really small (Pearson coef < 0.02). It is difficult to evaluate the impact of such correlations on downstream analyses and no one would ever consider seriously such low (spurious) correlations.

3. The authors should evaluate metacells qualities based on standard metrics used in the metacell field (e.g. compactness, separation, size distribution, purity, etc.). It would also be useful to project the metacells in the single-cell space to see whether metacells are well distributed over the single cell space.

4. In Table 1: the number of gene pairs with significant PPI enrichment for LSMetacell and the primary method are relatively close. The authors should test whether these differences are significant or not. Statistics indicating the significance of the differences observed between the methods benchmarked should also be added to Figure 2.

5. The benchmark of LSMetacell against published methods is not comprehensive: well-known methods need to be included such as SEACells (Persad et al. Nature biotechnology, 2023) and SuperCell (Bilous et al. BMC bioinformatics, 2022).

6. In the implementation of LSMetacell (R package on github), it seems that there is no possibility to link a metacell to the single cells they contain. This is critical for the evaluation of metacells quality (including the metrics mentioned in the comment n°3).

7. This algorithm might be computationally heavy for large datasets. The authors should mention the resources needed to build metacells with LSMetacells on datasets with various sizes.

8. The authors applied LSMetacell to a brain dataset and identified biologically meaningful cell-type-specific co-expression networks. However, the authors do not show whether these networks could also be found using the algorithms that do not consider library size in the metacell construction (or directly from the single-cell data without considering metacells). Here again, much more extensive benchmark is needed

9. In the discussion, the authors mention that one of the limitation of the method is that cell types have to be defined before using LSMetacells to build metacells, this point should be emphasized in the method section. It was indeed not clear to me that metacells should be built within each cell type.

10. On GitHub the example of the readme file cannot be run, the “dpfc” object is not defined. The authors should consider using publicly available dataset such as the Seurat pbmc data.

11. The quality of the figures needs to be substantially improved. Many of them are just unreadable due ot catastrophic choices of font sizes, axis names,… As reviewer, we spend lots of time evaluating papers, so a minimal efforts to make Figures readable and consistent would be much appreciated

**Have the authors made all data and (if applicable) computational code underlying the findings in their manuscript fully available?**

Reviewer #1: Yes

Reviewer #2: Yes

PLOS authors have the option to publish the peer review history of their article (what does this mean?). If published, this will include your full peer review and any attached files.

Reviewer #1: No

Reviewer #2: No

**Figure resubmission:**

**Reproducibility:**



---

## [Decision Letter · Decision Letter 2]

19 Oct 2025

PCOMPBIOL-D-25-00872R2

Library Size-Stabilized Metacells Construction Enhances Co-Expression Network Analysis in Single-Cell Data

PLOS Computational Biology

Dear Dr. Zhu,

Thank you for submitting your manuscript to PLOS Computational Biology. After careful consideration, we feel that it has merit but does not fully meet PLOS Computational Biology's publication criteria as it currently stands. Therefore, we invite you to submit a revised version of the manuscript that addresses the points raised during the review process.

Please submit your revised manuscript within 10 days Dec 19 2025 11:59PM. If you will need more time than this to complete your revisions, please reply to this message or contact the journal office at ploscompbiol@plos.org. Please include the following items when submitting your revised manuscript:

We look forward to receiving your revised manuscript.

Kind regards,

Marcel Holger Schulz, Ph.D.

Academic Editor

PLOS Computational Biology

Ilya Ioshikhes

Section Editor

PLOS Computational Biology

**Additional Editor Comments :**

The reviewers have no conceptual problems after the last revision. However, there were concerns with the quality of the figures. Please make sure that all figures are vector graphics and are of high enough resolution for the main and supplemental figures. Also make sure that font sizes between subfigures are comparable. Currently there is quite a difference in font size comparing Figure 1,2 and 3. Also please rename the y-axis of Figure 2 as something easier to understand, such as Z-summary statistic

For Figure 3b, why's the CREAD score abbreviated as Ceradsc and not as Creadsc. Is this a typo?

Also in Figure 3B and 3C the names for modules are different, because in B its the eigen genes (MEyellow) and in C the modules (MC_yellow). Please add a description of this difference to the figure caption, such that readers that do not know WGCNA well can follow.

**Reviewers' comments:**

Reviewer's Responses to Questions

Reviewer #2: Most of my comments have been addressed, but I still have some remaining issues.

1) Figures are still of very low quality with many plots being highly pixelized (not only in the pdf, also in the tif files). No effort has been made to avoid large empty spaces, points being cut by axis (e.g., Figure 3c), axis names reflecting internal variable in the code (“Zsummary.pres”), lack of axis, etc., etc. I was very clear in my previous report that Figures needed to be improved, and it looks like the authors could not care less about this point. This is utterly disappointing.

2) Lines 186-188 The authors mention that “LSMetacell achieved the lowest coefficient of variation, indicating superior stability of the generated meta-cells”. I would clarify in the main text that the coefficient of variation was computed based on library size (as described in the supplementary file) and thus the stability statement refers to library size stability.

3) In the discussion, lines 386-388 the authors mention that “in addition to library size control, other factors such as high compactness and separation of metacells may also play important roles in obtaining biologically meaningful co-expression networks”. Metacells of good quality should have low compactness and high separation not high compactness and separation.

4) In response to my 4th comment in the first round of revision, the authors say that they completed Table 1 by performing a “paired Wilcoxon test to compare the number of significantly enriched PPI pairs between LSMetacell and each benchmark method across multiple cell types”. However, I see only one pvalue in the legend and not a pvalue for each comparison.

**Have the authors made all data and (if applicable) computational code underlying the findings in their manuscript fully available?**

Reviewer #2: Yes

PLOS authors have the option to publish the peer review history of their article (what does this mean?). If published, this will include your full peer review and any attached files.

Reviewer #2: No

**Figure resubmission:**
---

## [Editor Report · Decision Letter 3]

3 Nov 2025

Dear Dr. Zhu,

We are pleased to inform you that your manuscript 'Library Size-Stabilized Metacells Construction Enhances Co-Expression Network Analysis in Single-Cell Data' has been provisionally accepted for publication in PLOS Computational Biology.

Best regards,

Marcel Holger Schulz, Ph.D.

Academic Editor

PLOS Computational Biology

Ilya Ioshikhes

Section Editor

PLOS Computational Biology

---

## [Editor Report · Acceptance letter]

PCOMPBIOL-D-25-00872R3

Library Size-Stabilized Metacells Construction Enhances Co-Expression Network Analysis in Single-Cell Data

Dear Dr Zhu,

I am pleased to inform you that your manuscript has been formally accepted for publication in PLOS Computational Biology. Your manuscript is now with our production department and you will be notified of the publication date in due course.

With kind regards,

Anita Estes
